# Morphologically and physiologically diverse fruits of two *Lepidium* species differ in allocation of glucosinolates into immature and mature seed and pericarp

Said Mohammed[1,2]°, Samik Bhattacharya[1]°*, Matthias Alexander Gesing[3]°, Katharina Klupsch[4], Günter Theißen[4], Klaus Mummenhoff[1‡], Caroline Müller[3‡]

1 Department of Biology, Botany, University of Osnabrück, Osnabrück, Germany, 2 Department of Biology, Debre Birhan University, Debre Birhan, Ethiopia, 3 Faculty of Biology, Department of Chemical Ecology, Bielefeld University, Bielefeld, Germany, 4 Matthias Schleiden Institute/Genetics, Friedrich Schiller University Jena, Jena, Germany

° These authors contributed equally to this work.
‡ These authors also contributed equally to this work.
* Samik.bhattacharya@biologie.uni-osnabrueck.de

**Data Availability Statement:** All relevant data are within the paper and its Supporting Information files.

## Abstract

The morphology and physiology of diaspores play crucial roles in determining the fate of seeds in unpredictable habitats. In some genera of the Brassicaceae different types of diaspores can be found. *Lepidium appelianum* produces non-dormant seeds within indehiscent fruits while in *L. campestre* dormant seeds are released from dehiscent fruits. We investigated whether the allocation of relevant defence compounds into different tissues in different *Lepidium* species may be related to the diverse dispersal strategy (indehiscent and dehiscent) and seed physiology (non-dormant and dormant). Total glucosinolate concentration and composition were analysed in immature and mature seeds and pericarps of *L. appelianum* and *L. campestre* using high-performance liquid chromatography. Moreover, for comparison, transgenic RNAi *L. campestre* lines were used that produce indehiscent fruits due to silencing of *LcINDEHISCENCE*, the *INDEHISCENCE* ortholog of *L. campestre*. Total glucosinolate concentrations were lower in immature compared to mature seeds in all studied *Lepidium* species and transgenic lines. In contrast, indehiscent fruits of *L. appelianum* maintained their total glucosinolate concentration in mature pericarps compared to immature ones, while in dehiscent *L. campestre* and in indehiscent RNAi-*LcIND L. campestre* a significant decrease in total glucosinolate concentrations from immature to mature pericarps could be detected. Indole glucosinolates were detected in lower abundance than the other glucosinolate classes (aliphatic and aromatic). Relatively high concentrations of 4-methoxyindol-3-ylmethyl glucosinolate were found in mature seeds of *L. appelianum* compared to other tissues, while no indole glucosinolates were detected in mature diaspores of *L. campestre*. The diaspores of the latter species may rather depend on aliphatic and aromatic glucosinolates for long-term protection. The allocation patterns of glucosinolates correlate with the morpho-physiologically distinct fruits of *L. appelianum* and *L. campestre* and

**Funding:** This work was supported by grants from the Deutsche Forschungsgemeinschaft to K.M. (MU 1137/8-2) and G.T. (TH 417/6-2). The funders had no role in study design, data collection and analysis, decision to publish, or preparation of the manuscript.

**Competing interests:** The authors have declared that no competing interests exist.

may be explained by the distinct dispersal strategies and the dormancy status of both species.

## Introduction

For seed plants, fruit structures and corresponding dispersal strategies are life history traits of particular importance influencing plant fitness. The functional dispersal units bearing mature seed, the diaspores, can show a high structural diversity, which influences the successful establishment of species in their respective habitat [1, 2]. In several angiosperms including the Brassicaceae family, two major fruit morphs can be found in various genera, namely dehiscent and indehiscent fruits. Dehiscent fruits are the most common fruit morph among the Brassicaceae and are assumed to be the ancestral diaspore morph in this family [3]. These fruits open along a predetermined dehiscence zone at the pericarp upon maturity and release their seeds [4]. In contrast, in indehiscent fruits the pericarp envelopes the seeds even after dispersal, until it finally decomposes and only then releases the seeds. Both fruit types are associated with different dispersal strategies, i.e., dehiscent fruits may escape unfavourable conditions via long-distance dispersal [5], while indehiscent fruits may escape in time by fractional or delayed germination [6]. The diaspore morph and mode of seed dispersal should thus be crucial in determining the defence requirements under natural selection conditions. Dehiscent fruits expose the seeds upon maturity and thus may require better protection for seeds than for pericarps. In contrast, in indehiscent fruits, the pericarp needs to be provided with a higher defence than the enclosed seeds.

Indeed, plant defence compounds are not equally distributed within a plant but qualitatively and quantitatively differ between tissues and in addition also with ontogenetic stage [7, 8]. Defensive natural products are expected to be optimally distributed to protect tissues with high fitness values and a higher likelihood of being attacked with priority [9, 10], as proposed by the optimal defence theory [11]. Seeds and their pericarps are metabolically active, vulnerable tissues of high value. The diaspores can experience fluctuations in the abiotic and biotic subterranean environment in long-term natural seed banks. Thus, it is paramount to mobilise as well as to optimise the provisioning of defensive metabolites in the different tissues that contribute to the diaspores according to their ontogeny and anticipated exposure to natural threats.

Glucosinolates (GSLs) are specialised (or also called secondary) plant metabolites that are specific to the order Brassicales and play an important role in defence against various generalist herbivores and pathogens [12, 13]. GSLs consist of a β-D-glucose residue that is connected by a sulfur atom to a (*Z*)-N-hydroximinosulfate ester as well as a benzenic, aliphatic or indole side chain [14]. The major classes of GSLs are formed from different amino acid precursors which can be readily hydrolysed by myrosinases upon tissue disruption, leading to the release of different volatile toxic hydrolysis products, such as nitriles and isothiocyanates [15]. Furthermore, enzymatic hydrolysis of indole GSLs results in unstable products, which upon reacting with other metabolites can form physiologically active indole compounds that might play a significant role in plant defence [16]. The highest concentrations of GLSs can be found in reproductive parts such as flowers and seeds [17]. A recent study revealed the allocation of different GSLs within seeds and pericarps of dehiscent and indehiscent fruits of *Aethionema* species (Brassicaceae) [7]. In these species, seeds accumulated higher GLS concentrations when ripe and particularly indole GLSs differed in their distribution between seed and pericarp

depending on the fruit morph. However, it remained unclear if there is a relationship between dehiscence/indehiscence and the GSL distribution in the diaspores due to distinct selection pressures on these different morphs and whether changing the dehiscence genetically may affect GLS allocation.

The genus *Lepidium* L. (Brassicaceae) consists of more than 200 annual and perennial species found on all continents except Antarctica, and includes some obnoxious weeds like hairy white top (*Lepidium appelianum* Al-Shehbaz; also, known as globe-podded hoary cress) and field pepper weed (*Lepidium campestre* (L.) W.T. Aiton) [18, 19]. The ancestral dehiscent fruit character in *L. campestre* is controlled by a gene regulatory network that includes one of the valve margin identity genes (*LcINDEHISCENT*, *LcIND*), the *L. campestre* ortholog of the *Arabidopsis thaliana* gene *INDEHISCENT*. Fruit indehiscence evolved several times independently within *Lepidium* s.l. and is found, for example, in *L. appelianum* [20]. Moreover, the indehiscent fruits of *L. appelianum* bear seeds, which are physiologically non-dormant and germinate immediately after maturity upon suitable conditions [21]. In contrast, released seeds of dehiscent *L. campestre* remain physiologically dormant after maturity [22] with a potential to form long-term seed banks [23]. These morpho-physiological distinctions between the fruits of *Lepidium* offer an excellent model system to analyse the congruence between defence and life-history strategies in maximising diaspore fitness.

In this study, we aimed to investigate whether the differences in dispersal strategy and seed dormancy status between the two *Lepidium* species correspond to the allocation of total and individual GSLs in immature and mature seeds and pericarps. Furthermore, we explored whether the transgenic abolition of the dehiscence zone in *L. campestre* affects the GSL distribution in the diaspores using transgenic RNAi-*LcIND L. campestre*. Finally, we tested the longevity in the seedbank for the wild type plants of both species. We discuss the allocation of GSLs in the diaspores of *L. appelianum* and *L. campestre* in relation to their dispersal strategy and their natural seedbank persistence and dormancy cycle.

## Materials and methods

### Seed sources

Seeds of *Lepidium appelianum* (KM 1754; obtained from J Gaskin, USDA, Fremont County, Wyoming, USA) and wild type *L. campestre* (KM 96; obtained from Botanical Garden, University of Zürich) were collected from mass propagations in the Botanical Garden, Osnabrueck University, Germany, in 2014 to 2015. Seeds of the transgenic *Lepidium campestre* line RNAi-*IND*a (henceforth termed RNAi-*LcIND*), in which silencing of *LcINDEHISCENT* by RNAi is established, resulting in indehiscent fruits, were collected from plants cultivated at Friedrich Schiller University Jena; for details of cloning, transformation and plant cultivation procedures, see [24]).

### Plant cultivation and sample harvest

Mature plants were grown from seeds on sterilised rooting-media agar plates (0.043% Murashige & Skoog Medium basal salt mixture, Duchefa, Haarlem, Netherlands; 1% Agar; pH 7) for four days at 4 ˚C in darkness, followed by incubation in a growth chamber at 14 ˚C with 18 h daylight (155 µm s$^{-1}$ m$^{-2}$). After 15–20 days, the germinated seedlings were transferred to 0.5 l pots filled with a mixture of soil (Einheitserde, Einheitserdewerke Gebr. Patzer GmbH & Co. KG): autoclaved sand: perlite (7:2:1). All plants were cultivated under identical conditions (22::12 ˚C, 18::6 h day::night, 47% relative humidity). The plants started flowering three months after germination and produced seeds after another two months.

We followed the fruit developmental stages as described by Ferrandiz et al. [25] for *A. thaliana* to harvest our samples representing specific ontogenetic stages. The two analysed stages of fruit development that we harvested, immature and mature fruits, correspond to the stages 17b and 19/20, respectively, for fruit development in *A. thaliana*. Stage 17b refers to a green, fully developed fruit, whereas stage 19/20 stands for a fruit that turns brown and papery and which can be easily broken to diaspores in the case of dehiscent fruits. Immature fruits were collected 28 days after the beginning of flowering, while mature fruits were harvested 28 days later. About 25 fruits were collected separately from every individual plant ($n$ = 7 per plant, species and transgenic line), quickly frozen in liquid nitrogen, stored at -80 ˚C and freeze-dried for at least 36 h (Alpha 1–4 LSC, Martin Christ Gefriertrocknungsanlagen GmbH, Germany). Following freeze-drying, the fruit tissues (pericarp and seed) were separated manually and stored over dry silica gel until GLS analysis.

## Glucosinolate extraction and analysis

Approximately 15 mg of dried samples were weighed (on a precision scale ME36S, accuracy 0.001 mg; Sartorius AG, Goettingen, Germany), homogenised in a mill for 20 sec at 20 kHz (Retsch, MM301, Haan, Germany), and extracted three times with 80% methanol, adding 20 µl of *p*-hydroxybenzyl GSL (mature pericarps of *L. appelianum* and *L. campestre* wild type) or allyl GLS (all remaining tissues; GLSs from Phytoplan Diehm & Neuberger, Heidelberg, Germany) as internal standards at the first extraction. Extracts were centrifuged and the supernatants applied on diethylaminoethyl columns [Sephadex A25 (Sigma Aldrich, St. Louis, MO, USA) swelled in 0.5 M acetic acid buffer (pH 5)]. The columns were washed with deionised water and purified sulfatase was added (following Graser et al. [26]). After overnight incubation, the desulfo GSLs [27] were eluted from the columns with ultra-pure water and samples were analysed by high performance liquid chromatography (HPLC) coupled to diode-array ion detection (HPLC-1200 Series, Agilent Technologies, Inc., Santa Clara, CA, USA) equipped with a Supelcosil LC 18 column (Supelco, Bellefonte, PA, USA). For *L. campestre* wild type samples, a 3 µm, 150 × 3 mm column was used, and the gradient started with 5% methanol, held for 6 min and was then increased stepwise to 95% within 13 min with a final hold at 95% for 2 min, followed by a cleaning cycle. For all other samples (*L. appelianum* and RNAi-*LcIND L. campestre*), a 5 µm, 250 × 4.6 mm column was used, and the gradient started with 5% methanol, was held for 10 min and was then stepwise increased to 95% within 22 min with a final hold at 95% for 3 min, followed by a cleaning cycle. Retention times and UV spectra were used to identify (desulfo) GSLs after comparing them to those of purified standards (Phytoplan, Heidelberg, Germany; Glucosinolates.com, Copenhagen, Denmark) and confirmation of some GSLs to an in-house library. Peak areas at 229 nm were integrated and glucosinolates quantified by incorporating the response factors listed in the ISO 9167 [28] as well as sample dry weights.

## Seed bank burial and germination trials

Intact indehiscent fruits of *L. appelianum* and isolated seeds from dehiscent fruit of *L. campestre* were buried at a depth of 5 cm from the soil surface in mesh bags at the field experimental sites of Botanical Garden, Osnabrück University, which allowed sufficient biotic and abiotic interaction. In each seed bag, either 25 intact fruits of *L. appelianum* (each enclosing 1–2 seeds) or 25 seeds of *L. campestre* were enclosed. Altogether, 15 and 27 seed bags of *L. appelianum* and *L. campestre*, respectively, were buried in May 2016 and three random bags for each species were excavated after every three months for germination trials until May 2018. Due to

the unavailability of enough replicate samples, seed bank burial experiments were limited for one year for *L. appelianum* and were not tested for RNAi-*LcIND L. campestre*.

Fruits and seeds were retrieved from the excavated seed bags by washing three times in sterilised water. Twenty-five fruits or seeds (3x replicates) were placed on sterile Petri dishes lined with moistened filter paper and sealed. Entire fruit enclosing the seeds of *L. appelianum* were incubated at a temperature of 25/15 ˚C with 12/12 h light/dark regime (light intensity = ca. 100 µmol m$^{-2}$ s$^{-1}$), while isolated seeds of *L. campestre* were incubated at 18/12 ˚C with similar light conditions. Visible protrusion of the radicle was recorded after 28 days at the completion of germination [29].

## Statistical analysis

All statistical analyses and graphical evaluations were performed with R-Version 3.6 [30]. Two factorial general linear models (GLM) analysis of variance were performed following Shapiro-Wilk-tests of normality of data on the effects of ontogeny (mature and immature) and tissue type (pericarp and seed) on total GSL concentrations in *Lepidium appelianum*, *L. campestre*, and transgenic RNAi-*LcIND L. campestre*. Significant differences in total GSL concentration between the tissue types and ontogeny were further evaluated by Tukey's post-hoc analysis of the GLM.

To visualise differences in GSL composition in the fruit tissues of the two *Lepidium* species and the transgenic *L. campestre*, non-metric multidimensional scaling (NMDS; R-package: vegan) was performed using Kulczynski distance as a dissimilarity index on the normalised data. Normalisation of the data was performed by replacing zeros or missing values with very small random numbers (<0.0005) and then applying Wisconsin double standardisation. Two-dimensional ordination plots were generated to resolve the distinction of GSL composition between diaspores of different ontogenetic stages (immature vs mature) and tissue types (pericarp vs seed) in *Lepidium species*. Furthermore, permutational multivariate analysis of variance (PERMANOVA) was performed, using the adonis function (R-package: vegan) [31] with 100 permutations for each species to determine the effect of factors (ontogeny, mature and immature; tissue type, pericarp and seed; and the interaction) on the GSL composition in all investigated samples. Independent supervised classification and feature selection method (Random Forest) was performed with MetaboAnalyst [32] for each data set to determine the most discriminating indole GSLs resolving the ontogeny in PCA biplots.

## Results

### Morpho- physiological differences among mature diaspores of *Lepidium*

The mature infructescences profoundly differ in morphology between the two *Lepidium* species. While mature indehiscent fruits of *L. appelianum* contain 1–2 seeds enclosed within the bulbous papery pericarp (Fig 1A and 1B), fruits of *L. campestre* dehisce upon maturity to release two seeds by detaching of the two fruit valves from the replum (Fig 1C and 1D). Post-transcriptional silencing of *LcIND* in *L. campestre* by RNAi [24] did not alter the overall morphology of the fruits but transformed them to be indehiscent *via* the abolition of the dehiscence zone at the fruit valve margin; however, these fruits still contained two seeds (Fig 1E and 1F).

In the germination trials, more than 85% *L. appelianum* seeds germinated immediately after maturity with no sign of dormancy, irrespective of whether they were enclosed within the pericarp or manually released [21]. However, freshly harvested *L. campestre* seeds exhibited non-deep physiological dormancy upon maturity and only germinated after 12–16 weeks of natural ageing. Similarly, the transgenic RNAi-*LcIND L. campestre* showed non-deep

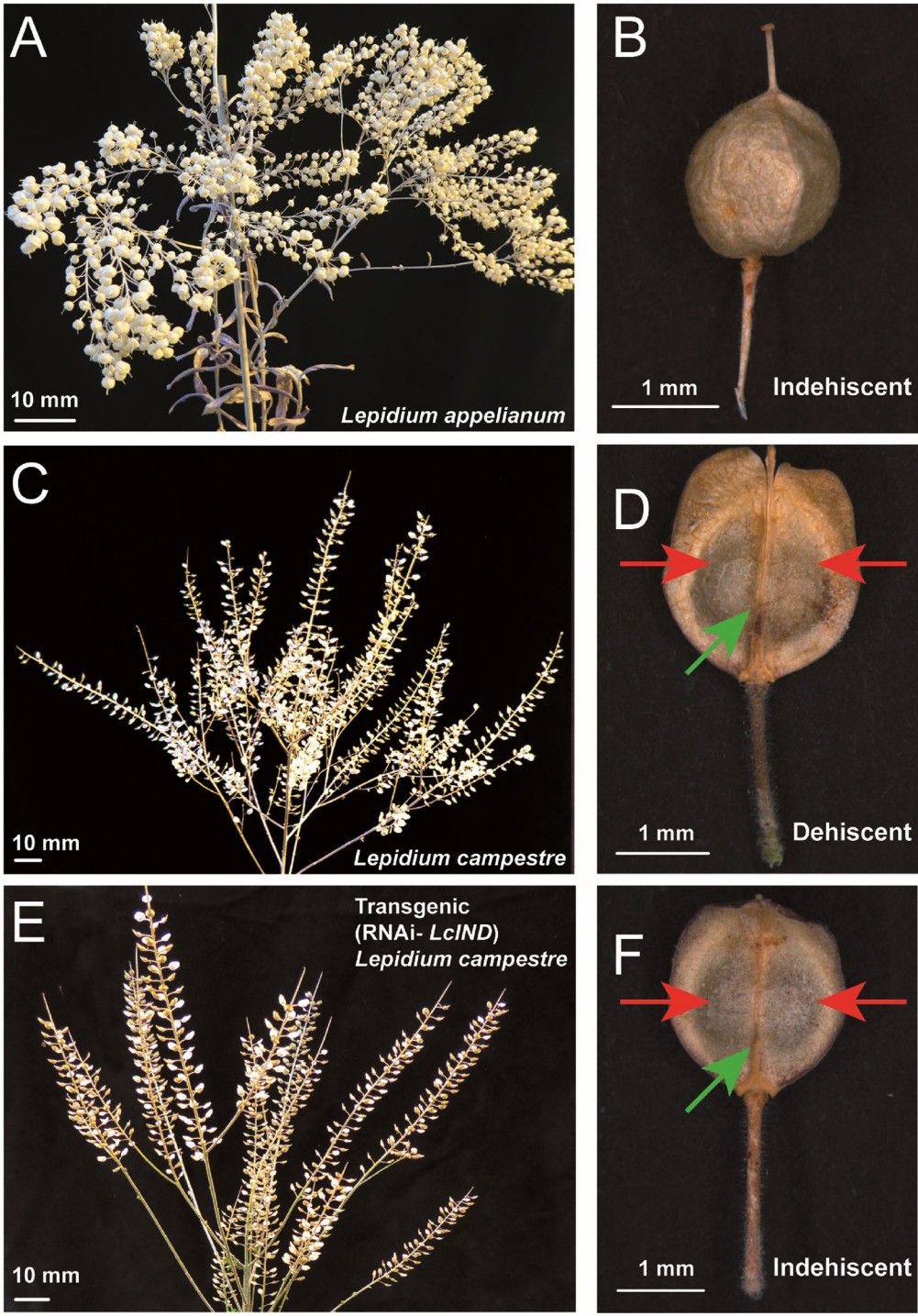

**Fig 1. Morphology of mature infructescences and fruits of *Lepidium appelianum*, *L. campestre*, and RNAi-*LcIND* L. campestre.** (A, B) Mature indehiscent fruits of *L. appelianum* contain 1–2 seeds enclosed within a bulbous papery pericarp. (C, D) Mature fruits of *L. campestre* dehisce upon maturity to release two seeds enclosed within the pericarp by detaching of the two fruit valves (red arrow) from the replum (green arrow). (E, F) Transgenic modification of *LcIND* in *L. campestre* produced indehiscent fruits containing two seeds enclosed within un-detached fruit valves from replum.

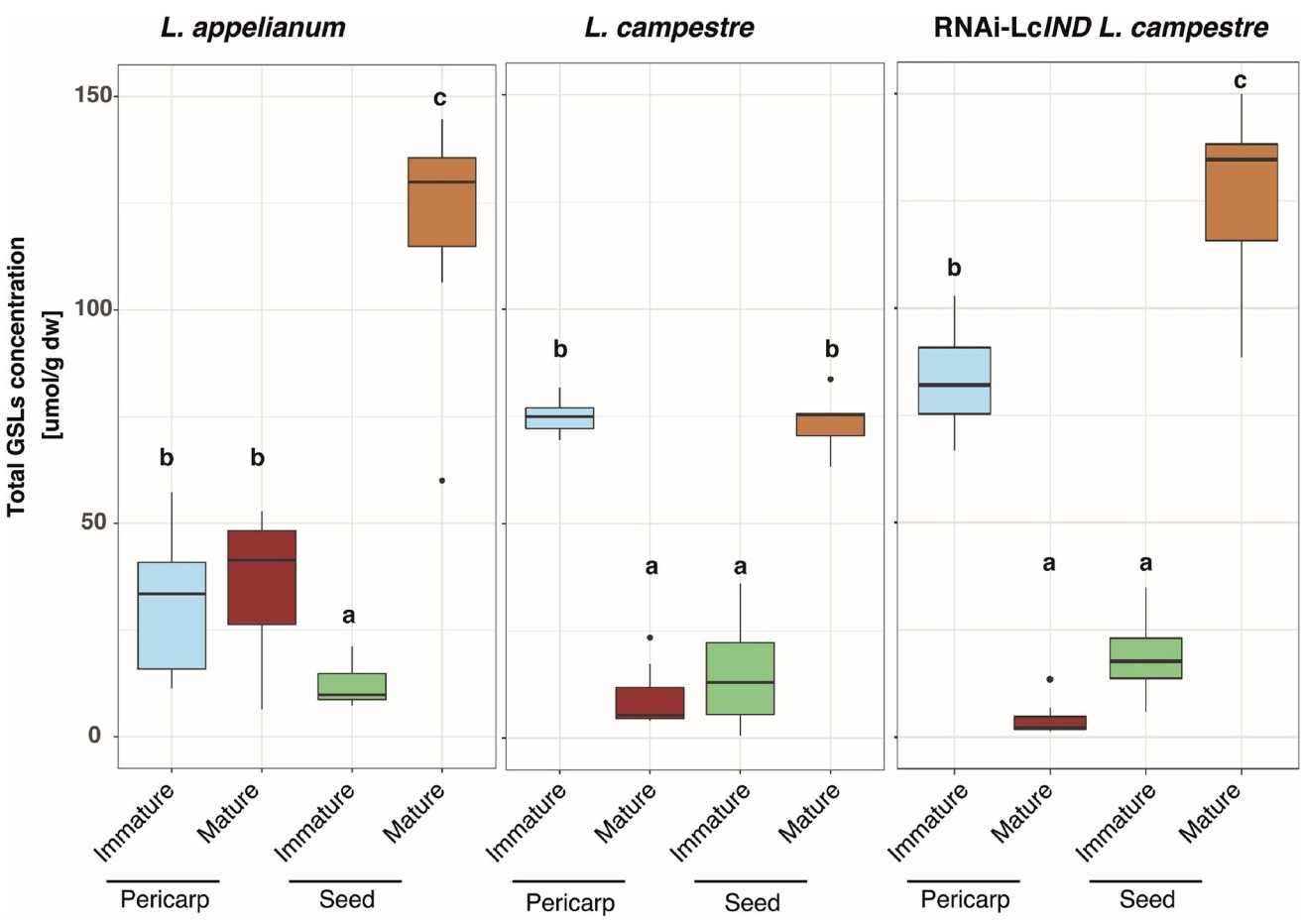

**Fig 2. Total glucosinolate concentrations differ among tissue types (pericarp vs seed) of immature and mature diaspores of *Lepidium appelianum*, *L. campestre*, and transgenic RNAi-*LcIND L. campestre*.** The concentrations of the total glucosinolates (µmol g$^{-1}$ DW) are displayed as box-whisker plots. Boxes show the median (line) as well as the 25$^{th}$ and 75$^{th}$ percentiles, whiskers extend to the 5$^{th}$ to 95$^{th}$ percentiles and dots indicate outliers, n = 7. Different letters within each plot indicate a significant difference in mean values in Tukey's post-hoc analysis following two factorial general linear models analyses of variance (see Table 1) on the effects of ontogeny (mature and immature) and tissue type (pericarp and seed) on total glucosinolate concentrations.

physiological dormancy of the seeds enclosed in the pericarp of indehiscent fruit, and germinated only after removing the seeds from the pericarp and after a similar period of natural ageing as in case of wild-type *L. campestre*.

## Mature functional diaspores of *Lepidium* contain high GSL concentrations

Significant differences in the total GSLs concentrations between tissues of different ontogenetic stages (immature and mature) and tissue types (pericarp and seed) were observed in both *Lepidium* species and in the transgenic RNAi-*LcIND L. campestre* (Fig 2, Table 1). The total GSLs concentrations did not differ significantly between immature and mature pericarps of *L. appelianum* ($\Delta_{mean}$ = 5 µmol g$^{-1}$, ANOVA, $F_{1,12}$ = 0.29, $P$ = 0.60). In contrast, generally high concentrations of GSLs in immature pericarps were considerably diminished on maturity in *L. campestre* ($\Delta_{mean}$ = -65 µmol g$^{-1}$, ANOVA, $F_{1,12}$ = 379.32, $P$< 0.001) and in RNAi-*LcIND L. campestre* ($\Delta_{mean}$ = -79 µmol g$^{-1}$, ANOVA, $F_{1,12}$ = 250.49, $P$< 0.001). The total GSL concentrations in mature seeds of all samples were significantly higher than in immature seeds, with

**Table 1. Two factorial general linear model analysis of variance on the effects of ontogeny (mature and immature) and tissue type (pericarp and seed) on total glucosinolate concentrations in *Lepidium appelianum* (n = 7), *L. campestre* (n = 7) and transgenic RNAi-*LcIND L. campestre* (n = 7).**

| Factors | ndf | *Lepidium appelianum* | | | *Lepidium campestre* | | | RNAi-*LcIND L. campestre* | | |
|---|---|---|---|---|---|---|---|---|---|---|
| | | ddf | F | P | ddf | F | P | ddf | F | P |
| Ontogeny (immature vs mature) | 1 | 24 | 36.4 | <0.001 | 23 | 0.24 | 0.626 | 24 | 8.65 | <0.01 |
| Tissue type (pericarp vs seed) | 1 | 24 | 1.75 | 0.198 | 23 | 0.16 | 0.686 | 24 | 30.77 | <0.001 |
| Ontogeny x Tissue type | 1 | 24 | 28.31 | <0.001 | 23 | 72.86 | <0.001 | 24 | 216.78 | <0.001 |

ndf = numerator degrees of freedom, ddf = denominator degrees of freedom.

particularly high differences in *L. appelianum* (10x), followed by mature seeds of RNAi-*LcIND L. campestre* (7x).

## Distinct patterns of glucosinolate distribution in *Lepidium* diaspores

The different GSLs were found in a distinct distribution in immature and mature pericarp and seeds of the indehiscent *L. appelianum* fruits (Fig 3). Apart from high concentrations of the benzenic GSL *p*-hydroxybenzyl GSL (p-OHB), the aliphatic GSLs 4-methylthio-3-butenyl GSL (4MSO3B) and 6-methylsulfinylhexyl GSL (6MSOH) and the indol GSL 4-methoxyindol-3-ylmethyl GSL were detected in all tissues of *L. appelianum*, whereas 4MTB was found in all tissues of this species except in mature pericarps. In contrast, in *L. campestre* and RNAi-*LcIND L. campestre*, only p-OHB, 5-methylsulfinylpentyl GSL (5MSOP) and 6MSOH were predominantly present, although with varying proportions, in all tissue types of different ontogenetic stages, while 4-methylsulfinylbutyl GSL (4MSOB) was present in all *L. campestre* tissues. While no indole GSLs were detectable in both immature and mature pericarps and the seed of *L. campestre*, traces of 4MOI3M were detected in immature pericarps and seeds of RNAi-*LcIND L. campestre*. In conclusion, some GSLs were species-specific for the two *Lepidium* species. While 4MSOB and 5MSOP were not detectable in *L. appelianum*, 4MSO3B and 4MTB were absent in wild type and transgenic *L. campestre*.

## Multivariate analyses resolve the distinction of glucosinolate composition between diaspores of different ontogenetic stages and tissue types in *Lepidium*

Non-metric multidimensional scaling (NMDS) 2D ordination plots resolved the GLS composition of different ontogenetic stages and tissue types in *Lepidium* (Fig 4). The NMDS plot of *L. appelianum* and *L. campestre* displayed clear dissimilarities between GSL compositions of immature and mature pericarps and seeds except for a partial overlap of ordinates between mature seeds and immature pericarps in *L. appelianum*. However, RNAi-*LcIND L. campestre* showed a weak distinction between the GLS compositions of immature pericarps and seeds but the GSL composition of the mature pericarps and seeds were well resolved.

Permutational multivariate analysis of variance (PERMANOVA) within each species revealed significant differences in GSL composition among tissue types, ontogeny and their interaction (Table 2). However, only the combined interaction between the GLS composition of tissue types and ontogeny contributed more to the dissimilarities within the species ($R^2$ = 0.31–0.67) than ontogeny ($R^2$ = 0.08–0.15) or tissue types ($R^2$ = 0.09–0.18) alone within each species.

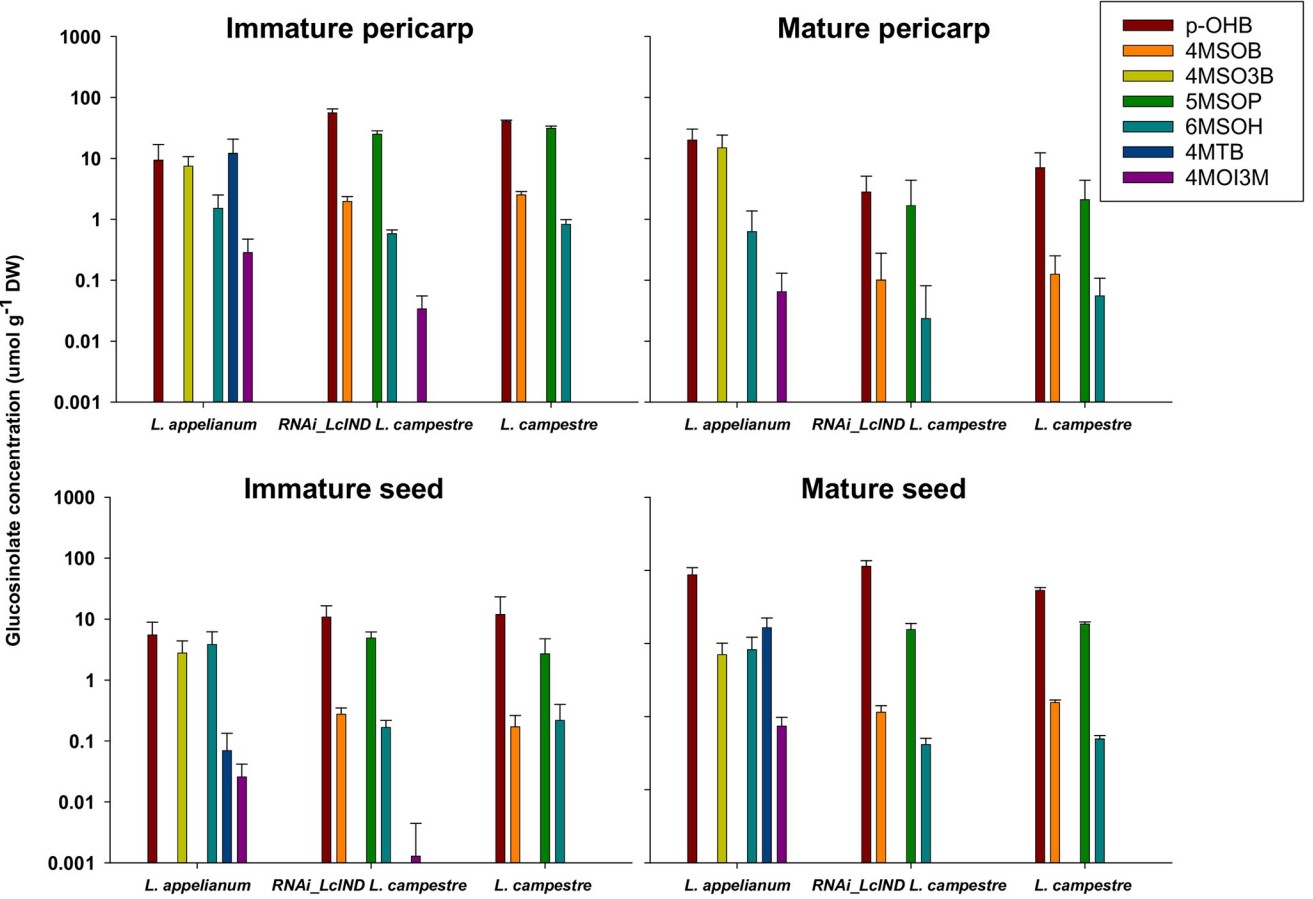

**Fig 3. Distinct patterns of glucosinolate distribution in *Lepidium* diaspores.** The mean concentrations (μmol/g DW, ±SE) of different glucosinolates (GSLs) measured in immature and mature pericarps and seeds of indehiscent (*L. appelianum*, transgenic- RNAi-*LcIND L. campestre*) and dehiscent (*L. campestre*) fruits are represented as column plots. The colour legends for the seven GSLs are indicated with the abbreviations: p-OHB, *p*-hydroxybenzyl GSL; 4MSOB, 4-methylsulfinylbutyl GSL; 4MSO3B, 4-methylsulfinyl-3-butenyl GSL; 5MSOP, 5-methylsulfinylpentyl GSL; 6MSOH, 6-methylsulfinylhexyl GSL; 4MTB, 4-methylthiobutyl GSL; 4MOI3M, 4-methoxy-indol-3-ylmethyl GSL.

## Distinct life-histories of *L. appelianum* and *L. campestre* explain diverse chemical defence strategies to survive in soil seedbank

Germination of seeds (enclosed in fruits = indehiscent) of physiologically non-dormant *L. appelianum* [21] increased from 85% (not buried fresh seeds) to 100% after three to six months of burial in the soil seed bank (Fig 5), and then declined gradually to less than 50% after 12 months of burial. The enclosed seeds were also gradually decayed by partial or full decomposition of pericarp leading to exposed seeds, out of which only 50% remained viable at the end of burial period. Conversely, only 50% of fresh, not buried seeds of physiologically dormant *L. campestre* germinated at the start of the seed bank burial experiment, and more than 80% germination was achieved after three months of burial. The germination percentage declined dramatically after nine months of burial to less than 50% germination, although the buried seeds remained viable and showed very little signs of decay. The natural dormancy cycle of *L. campestre* became evident from a gradual increase in germination percentage from 15–21 months after burial (Fig 5).

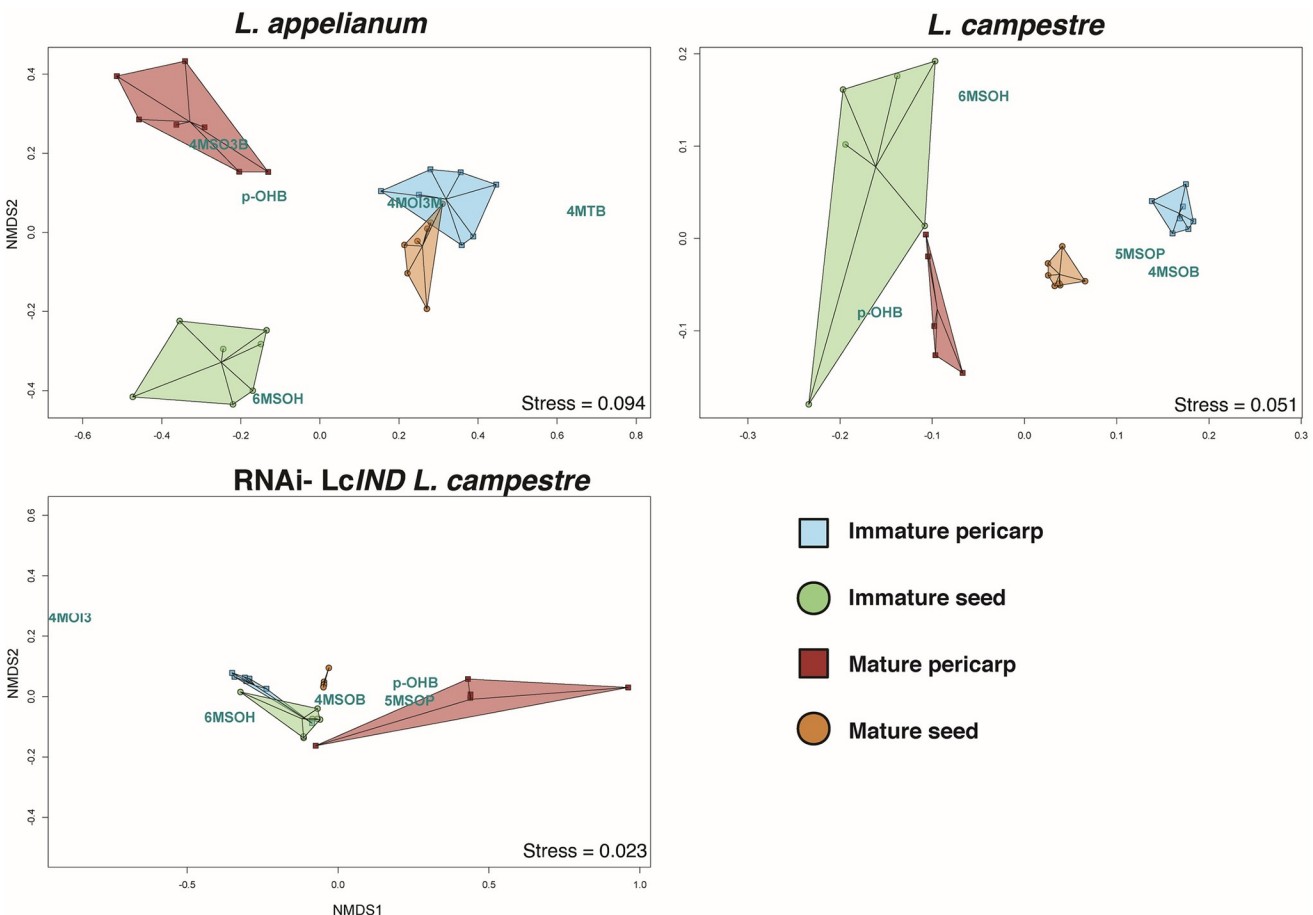

**Fig 4. Non-Metric Multidimensional Scaling (NMDS) 2D ordination plots resolve the distinction of glucosinolate composition between diaspores of different ontogenetic stages (immature vs mature) and tissue types (pericarp vs seed) in *Lepidium*.** The composition and concentration of glucosinolate (GSLs) were analysed to distinguish their clustering patterns among immature (shades of green polygons) and mature (shades of brown polygon) pericarps (squares) and seeds (circles). Biological replicates (n = 6–7) within each group were displayed as ordispiders, which are connected from their corresponding centroids, and the relative distance between the points represent the Kulczynski dissimilarity index computed from the chemical data transformed by Wisconsin double-standardisation method. The relative proximity of the GSLs (blue text) to the clusters signifies their potential resolving capacity within each plot. Stress values (shown in the bottom right of each plot) represent the overall resolution power of the NMDS analysis for each plot ($< 0.05$ = excellent, $<0.1$ = good).

**Table 2. Permutational multivariate analysis of variance (PERMANOVA) using Kulczynski distance matrices on the effects of ontogeny (mature and immature), and tissue type (pericarp and seed) on glucosinolate composition in *Lepidium appelianum*, *L. campestre* and transgenic RNAi-*LcIND L. campestre* (n = 6–7).**

| Factors | ndf | *Lepidium appelianum* | | | *Lepidium campestre* | | | RNAi-*LcIND L. campestre* | | |
|---|---|---|---|---|---|---|---|---|---|---|
| | | ddf | F | $R^2$ | ddf | F | $R^2$ | ddf | F | $R^2$ |
| Ontogeny (immature vs mature) | 1 | 24 | 9.78*** | 0.15 | 23 | 11.76** | 0.08 | 24 | 13.74*** | 0.13 |
| Tissue type (pericarp vs seed) | 1 | 24 | 10.55*** | 0.16 | 23 | 15.03*** | 0.09 | 24 | 19.29*** | 0.18 |
| Ontogeny x Tissue type | 1 | 24 | 19.87*** | 0.31 | 23 | 100.88*** | 0.67 | 24 | 48.04*** | 0.46 |

ndf = numerator degrees of freedom, ddf = denominator degrees of freedom.

Significance codes:

***, P<0.001;

**, P<0.01.

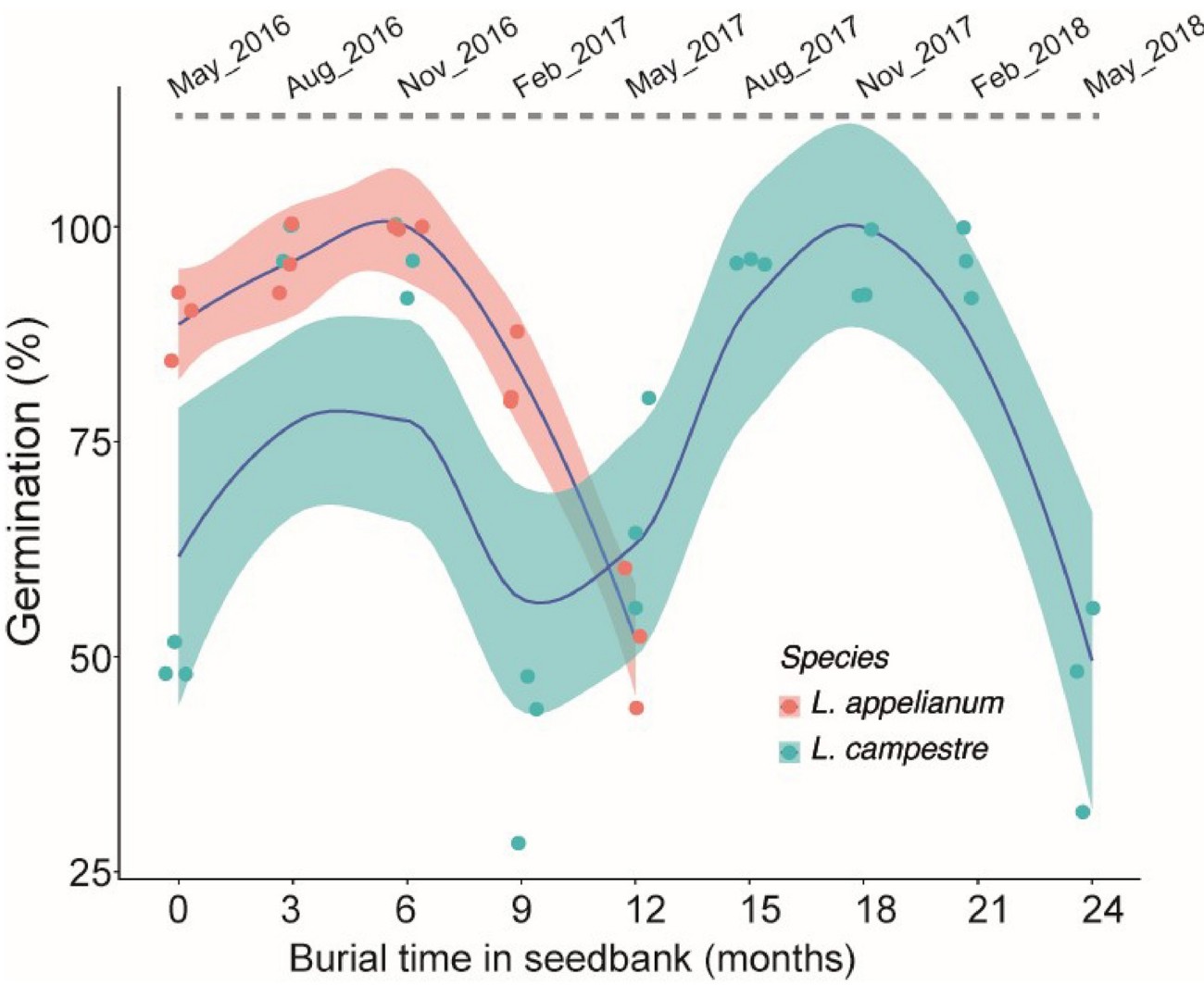

**Fig 5. Distinct dormancy cycling of indehiscent *L. appelianum* and dehiscent *L. campestre* in the soil seedbank.** Mean germination percentages of 75 seeds (from three randomly chosen seed bags) excavated every three months are represented as dot plots. The blue lines represent the regression over months of burial and germination with 95% confidences represented as shaded areas around the respective lines.

## Discussion

### Mature functional diaspores of *Lepidium* contain high GSL concentrations

The present study demonstrated a generally high concentration of GSLs in the immature pericarps irrespective of the species. Only in *L. appelianum*, the total GSLs remained similarly high in mature pericarps (Fig 2). These findings suggest a possible defensive role of GSLs in mature pericarps for the protection of indehiscent fruits of *L. appelianum*. In contrast, total GSL concentrations increased from immature to mature seeds in both species as well as in the transgenic line. These findings are in congruence with higher concentrations of GSLs found in mature seeds than in juvenile plant parts of other species within the genus, such as *L. peruvianum* [33] and *L. draba* [18]. Moreover, the consistently high concentrations of GSLs in seeds underpin their role in the protection of the tissue that will eventually give rise to the next generation irrespective of whether the seeds are enclosed within the papery thin pericarp of *L. appelianum* or released from the dehiscent fruits of *L. campestre*.

## Immature pericarp might act as source of all GSLs

Our results support the hypothesis that the immature pericarp in both fruit types (dehiscent and indehiscent) acts as a source of all GSLs and produces a comparable high level of GSLs, which are translocated to the seeds upon maturation [7]. Similarly, an increase in the accumulation of [35]S-labelled GSLs in the seeds of *Brassica napus* is correlated with a decrease in labelled GSLs in the pericarp [34]. Translocation of GSLs during seed maturation in dehiscent *A. thaliana* [35, 36], rather than *de novo* synthesis, can be explained from the analysis of GSL biosynthesis genes and corresponding transcription factors in *A. thaliana*, which revealed that seeds were unable to perform chain elongation and core biosynthesis steps of aliphatic GSLs, but were able to perform all the secondary modification steps on aliphatic GSLs [37]. However, for the investigated *Lepidium* species we cannot exclude the alternative explanation that GLSs are biosynthesized elsewhere and then transported to the pericarp and subsequently to the seeds.

## Total GSL concentrations differ among diaspores of *Lepidium* and are affected by Lc*IND* knockdown in *L. campestre*

While *L. campestre* showed a decrease in total GSL concentrations from immature to mature pericarp, *L. appelianum* showed no significant change from immature to mature pericarp in the present study. This does not by any means contradict the translocation hypothesis in the case of *L. appelianum*, since GSLs could have also been translocated from other plant tissues not investigated in the present study. Dehiscent fruits open their pericarp upon dispersal, so that the individual seeds are spread. Any defensive chemical left in the pericarps would mean an investment without fitness gain of the progeny, which would be in contrary to the optimal defence hypothesis [11]. Our data support the optimal defence theory, which states that plant defence compounds should be distributed in an optimal way to protect tissues with high fitness values and a higher probability of being attacked with priority [9, 10, 38].

The transgenic alteration of the dehiscence zone in RNAi-*LcIND L. campestre* yielded fruits resembling *L. campestre*, except that the functional dehiscence zone was absent (Fig 1). Nevertheless, the mature seeds in the transformed genotypes showed a higher increase of total GSLs compared to immature seeds than in the wild type plants. While there was no significant difference between the total GSL concentrations among the species, the GSL levels of *L. appelianum* and RNAi-*LcIND L. campestre* were higher than those in mature *L. campestre* seeds (S1 Fig). This observation suggests a preferred mobilisation of GSLs towards mature seeds than to mature pericarps according to the respective protective potentials. The allocation patterns may be further evaluated by overexpression of *IND* in *L. appelianum* to transform the fruit as dehiscent. Nevertheless, GSL concentrations were much higher in the mature pericarp of *L. appelianum* than in the transgenic *L. campestre*, suggesting that the genetic modification to produce indehiscent fruits only translated for the seeds to be protected more than the pericarp, a trait probably acquired over a long period of natural selection of released seeds from the dehiscent fruits of *L. campestre* [20, 24, 39].

## Distinct patterns of GSL distribution in *Lepidium* diaspores correlate with different potential selection pressures

While the functional diaspores of indehiscent *L. appelianum* (mature seed with pericarp) and dehiscent *L. campestre* (mature seed) are potentially optimally protected with a high amount of total GSLs (Fig 2), only mature seeds of transgenic indehiscent *L. campestre* were provisioned with high total GSLs but not their pericarps. However, mature pericarp and seeds of

*L. appelianum* showed a more diverse blend of GSLs than *L. campestre* and RNAi-*LcIND L. campestre*, signifying the readiness of the functional diaspore (pericarp with seed) against possible adversities during their stay in the seed bank (Fig 3). Moreover, the mature seeds and pericarps of *L. appelianum* contained an indole GSL, which was not detected in seeds or pericarps of *L. campestre* but was present in measurable concentrations in the immature tissues of RNAi-*LcIND L. campestre*. Over all samples, aromatic and aliphatic GSLs were more abundant than the indole GLS. Likewise, in other plants such as, for example, *A. thaliana*, indole glucosinolates are usually only present in low abundance (e.g., 4MOI3M; rosette leaves, 0.2–0.43; siliques, 0.01–0.02 μmol g$^{-1}$ DW; matured seeds, not detected) compared to at least tenfold higher concentrations of aliphatic glucosinolates (e.g., 4MSOB; rosette leaves, 2.5–10.6; siliques, 9.7–18.9; matured seeds, 2.43 μmol g$^{-1}$ DW) [42]. However, breakdown products of indole GSLs are known to be exceptionally potent as defensive compounds [16]. Furthermore, indole GSLs are readily inducible upon damage and can increase up to 20-fold [40]. The plants in the present study were not exposed to herbivory or pathogen damage, which may explain their rather low constitutive indole GSL contents. The most common GSL, p-OHB, was detected at consistently and comparatively higher concentration in mature seeds of *L. appelianum* and RNAi-*LcIND L. campestre* than in *L. campestre*, suggesting that the accumulation of this defence compound is variable between the indehiscent and the dehiscent fruits. Whether the genetic suppression of *LcIND* in RNAi-*LcIND L. campestre* correlates with the changes in the expression of GSL pathway genes needs further evaluation. In *A. thaliana*, *IND* regulates the auxin transport machinery in gynoecia and this phytohormone subsequently controls several biochemical pathways [41].

## Multivariate analyses resolve the distinction of GSL composition between diaspores of different ontogenetic stages and tissue types in *Lepidium*

In the multivariate NMDS analyses, the mature pericarps were separated from the other tissues in their GSL profiles only in *L. appelianum* and RNAi-*LcIND L. campestre* but not in *L. campestre*. Moreover, the high variation within the immature seeds of *L. campestre* suggests a high dissimilarity in GSL composition between the samples. Diversity in GSL patterns range from an individual scale with differences among plant tissues [42] to a within species scale with variation among individuals [43]. A high variation in GLS may have ecological consequences. For example, in *Brassica oleracea*, species richness and diversity of the herbivore community were found to be positively correlated with the length of the side chains in alkenyl GSLs [43]. A complex blend of GSLs among populations may make it difficult for herbivores to adapt to a specific GSL pattern. Expressing such high variation in the GLS profiles may be one reason for the global success of several invasive species such as *Bunias orientalis* [44]. The detected variation in GSL patterns among diaspores in the present study may likewise affect the interactions between *Lepidium* and their respective herbivore or pathogen communities.

Moreover, the myrosinase activity may differ between immature and mature seed and pericarp of different *Lepidium* species and influence the defensive potential of tissues. The myrosinase-GSL system is not only involved in defence against herbivores and pathogens but also in the sulfur and nitrogen metabolism and growth regulation of plants [45]. Further research on myrosinase activities and the distribution of other defence compounds apart from GSL in dehiscent and indehiscent fruits and their parts is needed. Although the morphological change from dehiscence to indehiscence in *L. campestre* altered the GSL profiles, the confounding effects of intricate genetic or biochemical pathways on the regulation of GSL profiles cannot be ruled out.

## Distinct life-histories of *L. appelianum* and *L. campestre* explain distinct chemical defence strategies to survive in soil seedbank

The distinct dormancy cycling of indehiscent *L. appelianum* and dehiscent *L. campestre* presumably requires different chemical defence strategies to survive in the soil seedbank. The seeds of *L. appelianum* are non-dormant and germinate immediately after maturation upon favourable conditions. In contrast, the released, dormant seeds of *L. campestre* require 3–6 months of mandatory after-ripening before full germination potential is achieved in the seedbank and tend to germinate in higher percentage in the following year [22, 23], i.e., 15–21 months after burial (Fig 5).

Therefore, in *L. campestre*, where dehiscent fruits expose the seeds upon maturity, seeds requires high provisioning of defensive compounds to survive in the seedbank and to germinate after this long period. On the other hand, the chemical protection provided by the papery thin persistent pericarp of *L. appelianum*, where the indehiscent fruit enclosed the seeds, cannot be overlooked in the scenario of their eventual persistence in the seedbank in the event of unfavourable conditions forcing the seeds to remain non-germinated despite their readiness to germinate immediately. Indeed, the unpredictable and harsh natural habitat conditions of *L. appelianum* in central and western Asia [46] often do not assure favourable conditions, compelling the seeds to remain potentially exposed to soil-inhabiting herbivores for an extended period of time. In the soil, the diaspores are, for example, exposed to plant-parasitic nematodes, which feed on plant roots as well as seeds [47, 48]. The nematicidal activity of GSLs and their degradation products may protect plant tissues against potential nematode infestation [49]. An elaborate seedbank analysis of all samples may reveal further insights into the fitness effects of the differential GSL allocation in the mature diaspores, given that the technical limitations barred us from testing the seedbank behaviour of *L. appelianum* for a longer period and of RNAi-*LcIND L. campestre*.

## Conclusion

The present study demonstrates that, although the GSL composition may differ among different Brassicaceae species, an overall trend to potentially translocate GSLs from less valuable tissues to the highly valuable reproductive tissue [35, 36] can be observed. Moreover, the findings of this study support a relation between GLS allocation in the different tissues and fruit morphs of *Lepidium* and the potential threats they are facing. The GLS allocation is also in congruence with the life-strategies and the long-term seed bank persistence of the morphologically and physiologically diverse *Lepidium* species. Ultimately, more research is needed to disentangle the potential genetic relationships between fruit morphology and biosynthesis of chemical defences.

## Supporting information

**S1 Fig. Total glucosinolate concentrations among tissue types (pericarp vs seed) of immature and mature diaspores of *Lepidium appelianum*, *L. campestre*, and transgenic RNAi-LcIND *L. campestre*.** The concentrations of the total glucosinolates ($\mu$mol $g^{-1}$ DW) are displayed as box-whisker plots. Boxes show the median (line) as well as the 25th and 75th percentiles, whiskers extend to the 5th to 95th percentiles and dots indicate outliers, n = 7 per species and line. Different letters within each plot indicate a significant difference in mean values in Tukey's post-hoc analysis following ANOVA on the effects of species on total glucosinolate concentrations.
(TIF)

## Acknowledgments

We thank Katharina Thomis and the gardeners of the Botanical Garden (Osnabrueck University) for plant propagation and help with seed bank experiments; and Rudolf Grupe and Ulrike Coja for their skilful technical assistance.

## Author Contributions

**Conceptualization:** Samik Bhattacharya, Günter Theißen, Klaus Mummenhoff, Caroline Müller.

**Data curation:** Said Mohammed, Samik Bhattacharya, Matthias Alexander Gesing, Katharina Klupsch, Günter Theißen, Klaus Mummenhoff, Caroline Müller.

**Formal analysis:** Said Mohammed, Samik Bhattacharya, Matthias Alexander Gesing, Katharina Klupsch, Günter Theißen, Klaus Mummenhoff, Caroline Müller.

**Funding acquisition:** Günter Theißen, Klaus Mummenhoff.

**Investigation:** Said Mohammed, Samik Bhattacharya, Matthias Alexander Gesing, Günter Theißen, Klaus Mummenhoff, Caroline Müller.

**Methodology:** Said Mohammed, Samik Bhattacharya, Matthias Alexander Gesing, Katharina Klupsch, Günter Theißen, Klaus Mummenhoff, Caroline Müller.

**Project administration:** Günter Theißen, Klaus Mummenhoff.

**Resources:** Günter Theißen, Klaus Mummenhoff, Caroline Müller.

**Software:** Samik Bhattacharya.

**Supervision:** Günter Theißen, Klaus Mummenhoff, Caroline Müller.

**Validation:** Samik Bhattacharya.

**Visualization:** Samik Bhattacharya.

**Writing – original draft:** Said Mohammed, Samik Bhattacharya, Matthias Alexander Gesing, Günter Theißen, Klaus Mummenhoff, Caroline Müller.

**Writing – review & editing:** Samik Bhattacharya, Günter Theißen, Klaus Mummenhoff, Caroline Müller.

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
