## [Decision Letter · Decision Letter 0]

27 Apr 2020

PONE-D-19-35159

Morphologically and physiologically diverse fruits of two *Lepidium* species differ in allocation of glucosinolates into immature and mature seed and pericarp

PLOS ONE

Dear Dr. Bhattacharya,

Thank you for submitting your manuscript to PLOS ONE. After careful consideration, we have decided that your manuscript does not meet our criteria for publication and must therefore be rejected.

I am sorry that we cannot be more positive on this occasion, but hope that you appreciate the reasons for this decision.

Yours sincerely,

Yong Pyo Lim

Academic Editor

PLOS ONE

Reviewers' comments:

Reviewer's Responses to Questions

**Comments to the Author**

1. Is the manuscript technically sound, and do the data support the conclusions?

Reviewer #1: No

Reviewer #2: Yes

2. Has the statistical analysis been performed appropriately and rigorously? 

Reviewer #1: No

Reviewer #2: Yes

3. Have the authors made all data underlying the findings in their manuscript fully available?

Reviewer #1: No

Reviewer #2: Yes

4. Is the manuscript presented in an intelligible fashion and written in standard English?

Reviewer #1: No

Reviewer #2: Yes

5. Review Comments to the Author

Reviewer #1: This paper has already been published in another journal.

Do not submit articles in multiple journals at the same time.

This can cause various problems in the future, including self-plagiarism.

Reviewer #2: This work is an interesting study regarding to make a link between molecular and physiological evidence on seed development and its ecological significance. However, there are still important paucity regarding the aim, questions, discussion and also figures preparation:

Abstract: lines 25-28: it is not clear, what is not known, what is the aim, question or aim of this work?‎

Introduction: lines 57-59: the relevance of defense compounds with the dehiscence/indehiscence ‎character is not justified. Authors need to better interpret the relationship between these two ‎parameters in the Introduction and Discussion, as well.‎

Lines 85-87: the question : if change in the dehiscence/indehiscence may be affect the GLs ‎distribution? Could be stated better: If there is a relationship between these two parameters? If yes, ‎what evolutionary and ecological criterion could be responsible for this relationship?‎

Results: the quality of Fig. 5 is not acceptable, I don’t mean the resolution, that is very low for all ‎figures. Fig. 4: the difference in the color (green) between immature and mature data is too low, could ‎not be recognized. Is this fig. necessary? If these data were obtained from the same seed collection, it ‎is redundant with Fig. 3, please remove it (fig 3 is adequate). ‎

Discussion: very long paragraphs. For example, the first paragraph could be divided at lines 361-406 ‎into two paragraphs, or much more exact division, with relevant subtitles, in order to make the text ‎readable.‎

I think study of only one parameter (GLs distribution) is not sufficient to establish a relationship ‎between the expression of a developmental gene and its function. Firstly, it would be very meaningful ‎when the RNAi-IND Arabidopsis plants were produced and its effect were experimented on the ‎dehiscence and GLs, in addition to Lepidium. Secondly, the authors must describe the type of the InD ‎gene (a bHLH gene), and discuss about the probable effect on the GSL-related biosynthetic or ‎metabolic gene, regarding their upstream elements, thus, a simple correlation is not sufficient as an ‎argument for the effect of IND on GSL biosynthesis an distribution, more molecular evidence should ‎be discussed for argumentation of the work. In the previous work of authors (Lenser T, Theissen G. ‎‎2013), the effect of RNAi-IND was observed on some of other developmental genes. What is the ‎importance of these effects on the observed data in the current work? It seems that, there is a gap ‎between two works, between two set of data: seed developmental genes and GSL metabolism and ‎distribution and the related genes. Please make clear. ‎

6. PLOS authors have the option to publish the peer review history of their article (what does this mean?). If published, this will include your full peer review and any attached files.

Reviewer #1: No

Reviewer #2: No

- - - - -

---

## [Author Response · Author response to Decision Letter 0]

29 May 2020

Comments to the Author

1. Is the manuscript technically sound, and do the data support the conclusions?

Reviewer #1: No

Reviewer #2: Yes

2. Has the statistical analysis been performed appropriately and rigorously? 

Reviewer #1: No

Reviewer #2: Yes

3. Have the authors made all data underlying the findings in their manuscript fully available?

Reviewer #1: No

Reviewer #2: Yes

4. Is the manuscript presented in an intelligible fashion and written in standard English?

Reviewer #1: No

Reviewer #2: Yes

As Reviewer #1 responded negatively to all of the above four questions contrary to the positive evaluation by Reviewer #2, we will be happy to receive some constructive feedback from Reviewer #1 regarding his/her evaluation. 

5. Review Comments to the Author

Reviewer #1: This paper has already been published in another journal.

Do not submit articles in multiple journals at the same time.

This can cause various problems in the future, including self-plagiarism.

We believe this is a misunderstanding as also confirmed by the senior editor, Jamie Males following our appeal request (see attached documents of previous communications regarding this matter). Therefore, we move forward to address the comments of the second reviewer.

Reviewer #2: This work is an interesting study regarding to make a link between molecular and physiological evidence on seed development and its ecological significance. However, there are still important paucity regarding the aim, questions, discussion and also figures preparation:

Abstract: lines 25-28: it is not clear, what is not known, what is the aim, question or aim of this work?‎

We tried to rephrase the aim and question of this work in a clear way in the revised version (lines 25-30).

Introduction: lines 57-59: the relevance of defense compounds with the dehiscence/indehiscence ‎character is not justified. Authors need to better interpret the relationship between these two ‎parameters in the Introduction and Discussion, as well.‎

We elaborated the introduction (lines 59-63) and discussion (lines 479-485, included section headers and conclusion) about the relevance of defence compounds with the specific seed dispersal characters (e.g. dehiscence/ indehiscence). We established in our previous publication (Plant Cell and Environment 42, 1381-1392, 2019) that defence compounds in Aethionema are concentrated in the tissues/diaspores according to their defence requirements under natural conditions. Seed dispersal mechanisms play an important role in determining such defence requirements. Dehiscent fruits expose the seeds upon maturity and thus require better protection for seeds than for pericarps against potential natural enemies. In contrast, in indehiscent fruits, the pericarp needs to be provided with an enhanced defence than the enclosed seeds. 

Lines 85-87: the question: if change in the dehiscence/indehiscence may be affect the GLs ‎distribution? Could be stated better: If there is a relationship between these two parameters? If yes, ‎what evolutionary and ecological criterion could be responsible for this relationship?‎

We thank the reviewer for this critical comment and we rephrased the question to ‘if there is a relationship between dehiscence/indehiscence and the GSL distribution in the diaspores’. The hypothesis that these traits may be correlated was derived from our findings in Aethionema, which we published earlier (Plant Cell and Environment 42, 1381-1392, 2019). In the present manuscript, we explored if similar patterns exist in another Brassicaceae species, Lepidium. To test the hypothesis, we chose L. appelianum (indehiscent fruit), L. campestre (dehiscent fruit), and genetically transformed L. campestre, which produce indehiscent fruits. While the results are not as dramatic as in our previous findings in Aethionema on the GSL distributions, we discussed the possible reasons of our results in the ecological context of the two members of Brassicaceae (Lepidium and Aethionema). We included more clarification in the revised manuscript.

Results: the quality of Fig. 5 is not acceptable, I don’t mean the resolution, that is very low for all ‎figures. 

We apologize for the low-resolution figures available to the reviewers, however we provided high resolution figures as outlined in the instructions to authors. Now we also appended good resolution figures at the end of the manuscript to overcome post-submission degradation of high-quality images incorporated in the manuscripts for the reviewers. 

We are not sure about what specific problem the reviewer is referring at in Fig 5 and why it is not acceptable. While the germination kinetics from seed bank burial experiments are typically expressed in similar graphs representing moving average plotting, we only appended the confidence interval as a cloud of 95% loess calculations. The graph is produced in the free statistical/graphical software R with utmost data resolution. Therefore, we want to keep the graph as it is now because it nicely depicts the dormancy cycling in the seed bank for the two species included in this study.

Fig. 4: the difference in the color (green) between immature and mature data is too low, could ‎not be recognized. Is this fig. necessary? If these data were obtained from the same seed collection, it ‎is redundant with Fig. 3, please remove it (fig 3 is adequate).

While the data used in both figures 3 and 4 are the same indeed, the analysis methods adapted for the different purposes of the two figures are distinct. The purpose of figure 3 was to show the details of the concentration of each individual GLS in different tissues and ontogenic levels for the three species and lines (L. appelianum, L. campestre and RNAi-LcIND L. campestre). However, figure 4 depicts the overall composition in GLSs and thus nicely demonstrates the difference among the tissue types across ontogeny based on a multivariate data analysis (NMDS). Therefore, we prefer to keep both figures, but can move Fig. 3 to the Supplement. We modified the colour scheme in Figures 2 and 4 to allow distinguishing the groups more clearly.

 ‎

Discussion: very long paragraphs. For example, the first paragraph could be divided at lines 361-406 ‎into two paragraphs, or much more exact division, with relevant subtitles, in order to make the text ‎readable.‎

We restructured the discussion with smaller and coherent parts, and included sub-heading as suggested by the reviewer for better readability.

I think study of only one parameter (GLs distribution) is not sufficient to establish a relationship ‎between the expression of a developmental gene and its function. Firstly, it would be very meaningful ‎when the RNAi-IND Arabidopsis plants were produced and its effect were experimented on the ‎dehiscence and GLs, in addition to Lepidium. Secondly, the authors must describe the type of the InD ‎gene (a bHLH gene), and discuss about the probable effect on the GSL-related biosynthetic or ‎metabolic gene, regarding their upstream elements, thus, a simple correlation is not sufficient as an ‎argument for the effect of IND on GSL biosynthesis an distribution, more molecular evidence should ‎be discussed for argumentation of the work. In the previous work of authors (Lenser T, Theissen G. ‎‎2013), the effect of RNAi-IND was observed on some of other developmental genes. What is the ‎importance of these effects on the observed data in the current work? It seems that, there is a gap ‎between two works, between two set of data: seed developmental genes and GSL metabolism and ‎distribution and the related genes. Please make clear. ‎

The reviewer is right in that we cannot provide a causal link between LcIND activity and GSL biosynthesis yet. However, for the indehiscent and dehiscent wild-type fruits there are good ecological explanations and selection pressures leading to evolution of different defence distributions. For the genetic modification of just the (developmental) indehiscence, one cannot expect that this is linked to GSL synthesis and the ecological and evolutionary drivers are lacking. In our manuscript, we used LcIND knockout plants just to see the effect of fruit indehiscence, whatever may have brought it about; we did not deal with a link between any specific transcription factor and GSL accumulation. As we have admitted already in our manuscript, this might develop into a more complicated story: "Whether the genetic suppression of LcIND in RNAi-LcIND L. campestre correlates with the changes in the expression of GSL pathway genes” needs further evaluation (line 439- 440). Here in this study, we claimed to explore only a façade of such complex correlation between dehiscence and GSLs, if there is any. In A. thaliana, IND regulates the auxin transport machinery in gynoecia and this phytohormone subsequently controls several biochemical pathways [Girin et. at. Plant Cell. 2011;23(10):3641-53]. Respective studies, e.g. employing comparative transcriptomics, are ongoing, but beyond the scope of this work here.

---

## [Decision Letter · Decision Letter 1]

8 Jul 2020

PONE-D-19-35159R1

Morphologically and physiologically diverse fruits of two Lepidium species differ in allocation of glucosinolates into immature and mature seed and pericarp

PLOS ONE

Dear Dr. Bhattacharya,

Thank you for submitting your manuscript to PLOS ONE. After careful consideration, we feel that it has merit but does not fully meet PLOS ONE’s publication criteria as it currently stands. Therefore, we invite you to submit a revised version of the manuscript that addresses the points raised during the review process.

We look forward to receiving your revised manuscript.

Kind regards,

Christophe Hano  and Juergen Kroymann

Academic Editors

PLOS ONE

Journal Requirements:

When submitting your revision, we need you to address these additional requirements.When submitting your revision, we need you to address these additional requirements.

1) Please amend your list of authors on the manuscript to ensure that each author is linked to an affiliation.

We note that you have included affiliation numbers 1,2,3 and #a, ¶ however affiliations 1,2,3 and #a, ¶, & have authors linked to them. Please add a text to affiliation & or remove if added in error.

2) Please remove your figures/ from within your manuscript file, leaving only the individual TIFF/EPS image files. These will be automatically included in the reviewer’s PDF

Additional comments from Editorial Board member:

a) In the abstract, the authors write 'Regarding the distribution of glucosinolate classes, high concentrations of 4-methoxyindol-3-ylmethyl glucosinolate [= 4MOI3M] were found in mature seeds of L. appelianum, while no indole glucosinolates were detected in mature diaspores of L. campestre.' (l. 37ff). But in Figure 3, the authors felt compelled to add extra panels for 4MOI3M 'due to its low abundance' (l. 294). The authors should clarify this, possibly by comparing with published values for 4MOI3M in Arabidopsis leaves or seeds (e.g., Brown et al., 2003, Phytochemistry 62, 471-481).

b) Fig. 3 could be improved by using a logarithmic scale for the y-axis and by putting values for L. appelianum, L. campestre and RNAi_LcIND L. campestre side-by-side for the different tissues and developmental stages.

c) In the discussion, the authors write 'These results support the hypothesis that the immature pericarp in both fruit types [...] acts as a source of all GSLs and produces a comparable high level of GSLs, which are translocated to the seeds upon maturation.' - Actually, it is striking that 4MOI3M is the only indole glucosinolate that is present in pericarps or seeds of L. appelianum and RNAi_LcIND L. campestre. Based on my knowledge of the pathways (Pfalz et al., 2009, Plant Cell 21, 985-999; Pfalz et al., 2011, Plant Cell 23, 716-729; Pfalz et al., 2016, Plant Physiol 172, 2190-2203) I would also expect the presence of indol-3-ylmethyl (I3M) and 4-hydroxy-indol-3ylmethyl (4OHI3M), the precursors of 4MOI3M, if the pericarp were producing indole glucosinolates. In fact, the absence of I3M and 4OHI3M in pericarp and seeds strongly suggests that 4MOI3M is biosynthesized elsewhere and then transported to pericarp and, subsequently, seeds.

Reviewers' comments:

Reviewer's Responses to Questions

**Comments to the Author**

1. If the authors have adequately addressed your comments raised in a previous round of review and you feel that this manuscript is now acceptable for publication, you may indicate that here to bypass the “Comments to the Author” section, enter your conflict of interest statement in the “Confidential to Editor” section, and submit your "Accept" recommendation.

Reviewer #2: All comments have been addressed

Reviewer #3: (No Response)

2. Is the manuscript technically sound, and do the data support the conclusions?

Reviewer #2: Yes

Reviewer #3: Partly

3. Has the statistical analysis been performed appropriately and rigorously? 

Reviewer #2: Yes

Reviewer #3: Yes

4. Have the authors made all data underlying the findings in their manuscript fully available?

Reviewer #2: Yes

Reviewer #3: Yes

5. Is the manuscript presented in an intelligible fashion and written in standard English?

Reviewer #2: Yes

Reviewer #3: No

6. Review Comments to the Author

Reviewer #2: The authors considered the majority of my comments in the manuscript, however, the response letter seems to be more convincing that the manuscript revised text regrading my questions. It means authors amended the text very conservatively and sparingly. Anyway, I think this paper is now suitable for publication, after improvement of the resolution of Fig. 3 that is still very low.

Reviewer #3: Mohammed et al described an interesting study toward explaining the different patterns of glucosinolates allocation into immature and mature seed and pericarp in morphologically and physiologically diverse fruits of two Lepidium species. They revealed that the concentration of glucosinolate not changed both in the immature and matured indehiscent pericarps of L. appelianum, while significant decrease in the dehiscent L.campestre and indehiscent RNAi-LcIND L. campestre. It is difficult to understand the link between pericarps dehiscent with glucosinolate concentration, and also the glucosinolate concentration and seeds dormancy. The allocation of different GSLs within seeds and pericarps of dehiscent and indehiscent fruits of Brassicaceae has been reported. Although there are no substantial flaws of the described work, its concept and methodology as well as the obtained results contain fewer exciting points.

7. PLOS authors have the option to publish the peer review history of their article (what does this mean?). If published, this will include your full peer review and any attached files.

Reviewer #2: No

Reviewer #3: No

---

## [Author Response · Author response to Decision Letter 1]

30 Jul 2020

Journal Requirements:

1) Please amend your list of authors on the manuscript to ensure that each author is linked to an affiliation. We note that you have included affiliation numbers 1,2,3 and #a, ¶ however affiliations 1,2,3 and #a, ¶, & have authors linked to them. Please add a text to affiliation & or remove if added in error.

We amended the list of authors with linked affiliations. Previously, we tried to follow the ‘instructions to authors’ for assigning the symbols suggested by the journal team (e.g., first set of equal contribution by ‘¶’, second set of equal contribution by ‘&’, current address of an author as ‘#a’, etc.). We recognized that the suggested system leads to more confusion than clarity. Therefore, we now used in the revised manuscript numbers for affiliations and superscripted symbols for other qualifiers (‘*’, corresponding author; ‘¶”, first set of equal contributions; ‘¶¶’, second set of equal contributions). 

2) Please remove your figures/ from within your manuscript file, leaving only the individual TIFF/EPS image files. These will be automatically included in the reviewer’s PDF

We removed the figures from the manuscript.

Additional comments from Editorial Board member:

a) In the abstract, the authors write 'Regarding the distribution of glucosinolate classes, high concentrations of 4-methoxyindol-3-ylmethyl glucosinolate [= 4MOI3M] were found in mature seeds of L. appelianum, while no indole glucosinolates were detected in mature diaspores of L. campestre.' (l. 37ff). But in Figure 3, the authors felt compelled to add extra panels for 4MOI3M 'due to its low abundance' (l. 294). The authors should clarify this, possibly by comparing with published values for 4MOI3M in Arabidopsis leaves or seeds (e.g., Brown et al., 2003, Phytochemistry 62, 471-481). 

We rephrased the corresponding sentences in the abstract to clarify that we detected generally low concentrations of indole glucosinolates in all analyzed fruit tissues (seeds, and pericarp) compared to aliphatic and aromatic glucosinolates (lines: 38-40). The term ‘relative high concentration of 4MOI3M in mature seeds’ in L. appelianum is only relevant for comparing among the other tissue types/ species. We added a statement on the abundance of the different GLS classes as well as the values of A. thaliana for comparison in the discussion (lines: 432-437).

b) Fig. 3 could be improved by using a logarithmic scale for the y-axis and by putting values for L. appelianum, L. campestre and RNAi_LcIND L. campestre side-by-side for the different tissues and developmental stages. 

We modified the figure 3 according to the suggestion by using logarithmic scale for the y-axis and by grouping the values of L. appelianum, RNAi-LcIND L. campestre, and L. campestre side by side.

c) In the discussion, the authors write 'These results support the hypothesis that the immature pericarp in both fruit types [...] acts as a source of all GSLs and produces a comparable high level of GSLs, which are translocated to the seeds upon maturation.' - Actually, it is striking that 4MOI3M is the only indole glucosinolate that is present in pericarps or seeds of L. appelianum and RNAi_LcIND L. campestre. Based on my knowledge of the pathways (Pfalz et al., 2009, Plant Cell 21, 985-999; Pfalz et al., 2011, Plant Cell 23, 716-729; Pfalz et al., 2016, Plant Physiol 172, 2190-2203) I would also expect the presence of indol-3-ylmethyl (I3M) and 4-hydroxy-indol-3ylmethyl (4OHI3M), the precursors of 4MOI3M, if the pericarp were producing indole glucosinolates. In fact, the absence of I3M and 4OHI3M in pericarp and seeds strongly suggests that 4MOI3M is biosynthesized elsewhere and then transported to pericarp and, subsequently, seeds.

Thank you for pointing this out. We added now a statement in the discussion: “However, for the investigated Lepidium species we cannot exclude the alternative explanation that GLSs are biosynthesized elsewhere and then transported to the pericarp and subsequently to the seeds” (lines: 389-391). We do not restrict this to the indole GLS, but rather keep the statement general, as this may be the case for all investigated GLSs.

Reviewers' comments:

Reviewer's Responses to Questions

Comments to the Author

1. If the authors have adequately addressed your comments raised in a previous round of review and you feel that this manuscript is now acceptable for publication, you may indicate that here to bypass the “Comments to the Author” section, enter your conflict of interest statement in the “Confidential to Editor” section, and submit your "Accept" recommendation.

Reviewer #2: All comments have been addressed

Reviewer #3: (No Response)

2. Is the manuscript technically sound, and do the data support the conclusions?

Reviewer #2: Yes

Reviewer #3: Partly

3. Has the statistical analysis been performed appropriately and rigorously? 

Reviewer #2: Yes

Reviewer #3: Yes

4. Have the authors made all data underlying the findings in their manuscript fully available?

Reviewer #2: Yes

Reviewer #3: Yes

5. Is the manuscript presented in an intelligible fashion and written in standard English?

Reviewer #2: Yes

Reviewer #3: No

6. Review Comments to the Author

Reviewer #2: The authors considered the majority of my comments in the manuscript, however, the response letter seems to be more convincing that the manuscript revised text regrading my questions. It means authors amended the text very conservatively and sparingly. Anyway, I think this paper is now suitable for publication, after improvement of the resolution of Fig. 3 that is still very low.

We have included more clarification in this revised version. We also reformatted the figure 3 following the suggestion of one of the reviewer and enhanced the resolution.

Reviewer #3: Mohammed et al described an interesting study toward explaining the different patterns of glucosinolates allocation into immature and mature seed and pericarp in morphologically and physiologically diverse fruits of two Lepidium species. They revealed that the concentration of glucosinolate not changed both in the immature and matured indehiscent pericarps of L. appelianum, while significant decrease in the dehiscent L.campestre and indehiscent RNAi-LcIND L. campestre. It is difficult to understand the link between pericarps dehiscent with glucosinolate concentration, and also the glucosinolate concentration and seeds dormancy. The allocation of different GSLs within seeds and pericarps of dehiscent and indehiscent fruits of Brassicaceae has been reported. Although there are no substantial flaws of the described work, its concept and methodology as well as the obtained results contain fewer exciting points.

We elaborated the introduction (lines: 58-65) and discussion (lines: 398-404, 486-498) about the relevance of defence compounds with the specific seed dispersal characters (e.g. dehiscence/ indehiscence). We established in our previous publication (Plant Cell and Environment 42, 1381-1392, 2019) that defence compounds in Aethionema are concentrated in the tissues/diaspores according to their defence requirements under natural conditions. Seed dispersal mechanisms play an important role in determining such defence requirements. Dehiscent fruits expose the seeds upon maturity and thus require better protection for seeds than for pericarps against potential natural enemies. In contrast, in indehiscent fruits, the pericarp needs to be provided with an enhanced defence than the enclosed seeds. 

In the present manuscript, we explored if similar relationship between dehiscence/indehiscence and the GSL distribution in the diaspores exist in another Brassicaceae species, Lepidium. To test the hypothesis, we chose L. appelianum (indehiscent fruit), L. campestre (dehiscent fruit), and genetically transformed L. campestre, which produce indehiscent fruits. While the results are not as dramatic as in our previous findings in Aethionema on the GSL distributions, we discussed the possible reasons of our results in the ecological context of the two members of Brassicaceae (Lepidium and Aethionema).

---

## [Decision Letter · Decision Letter 2]

12 Aug 2020

Morphologically and physiologically diverse fruits of two Lepidium species differ in allocation of glucosinolates into immature and mature seed and pericarp

PONE-D-19-35159R2

Dear Dr. Bhattacharya,

We’re pleased to inform you that your manuscript has been judged scientifically suitable for publication and will be formally accepted for publication once it meets all outstanding technical requirements.

Kind regards,

Christophe Hano

Academic Editor

PLOS ONE

Additional Editor Comments (optional):

Reviewers' comments:

Reviewer's Responses to Questions

**Comments to the Author**

1. If the authors have adequately addressed your comments raised in a previous round of review and you feel that this manuscript is now acceptable for publication, you may indicate that here to bypass the “Comments to the Author” section, enter your conflict of interest statement in the “Confidential to Editor” section, and submit your "Accept" recommendation.

Reviewer #2: All comments have been addressed

Reviewer #3: All comments have been addressed

2. Is the manuscript technically sound, and do the data support the conclusions?

Reviewer #2: Partly

Reviewer #3: Yes

3. Has the statistical analysis been performed appropriately and rigorously? 

Reviewer #2: Yes

Reviewer #3: Yes

4. Have the authors made all data underlying the findings in their manuscript fully available?

Reviewer #2: Yes

Reviewer #3: Yes

5. Is the manuscript presented in an intelligible fashion and written in standard English?

Reviewer #2: Yes

Reviewer #3: Yes

6. Review Comments to the Author

Reviewer #2: The text and particularly Discussion could be better, but any way, I think it is acceptable at this form.

Reviewer #3: The authors have addressed all my concerns in the revised MS. I recommend publishing the manuscript.

7. PLOS authors have the option to publish the peer review history of their article (what does this mean?). If published, this will include your full peer review and any attached files.

Reviewer #2: No

Reviewer #3: No

---

## [Editor Report · Acceptance letter]

13 Aug 2020

PONE-D-19-35159R2 

Morphologically and physiologically diverse fruits of two Lepidium species differ in allocation of glucosinolates into immature and mature seed and pericarp 

Dear Dr. Bhattacharya:

I'm pleased to inform you that your manuscript has been deemed suitable for publication in PLOS ONE. Congratulations! Your manuscript is now with our production department. 

Kind regards, 

on behalf of

Dr. Christophe Hano 

Academic Editor

PLOS ONE